# LIBERO: Benchmarking Knowledge Transfer for Lifelong Robot Learning

[†]**Bo Liu**,[\*] [†]**Yifeng Zhu**[\*], [‡]**Chongkai Gao**[\*], [†]**Yihao Feng**
[†]**Qiang Liu,** [†]**Yuke Zhu,** [†,§]**Peter Stone**
[†]The University of Texas at Austin, [§]Sony AI, [‡]Tsinghua University
{bliu,yifengz,lqiang,yukez,pstone}@cs.utexas.edu
yihao.ac@gmail.com, gck20@mails.tsinghua.edu.cn

## Abstract

Lifelong learning offers a promising paradigm of building a generalist agent that learns and adapts over its lifespan. Unlike traditional lifelong learning problems in image and text domains, which primarily involve the transfer of declarative knowledge of entities and concepts, lifelong learning in decision-making (LLDM) also necessitates the transfer of procedural knowledge, such as actions and behaviors. To advance research in LLDM, we introduce LIBERO, a novel benchmark of lifelong learning for robot manipulation. Specifically, LIBERO highlights five key research topics in LLDM: **1)** how to efficiently transfer declarative knowledge, procedural knowledge, or the mixture of both; **2)** how to design effective policy architectures and **3)** effective algorithms for LLDM; **4)** the robustness of a lifelong learner with respect to task ordering; and **5)** the effect of model pretraining for LLDM. We develop an extendible *procedural generation* pipeline that can in principle generate infinitely many tasks. For benchmarking purpose, we create four task suites (130 tasks in total) that we use to investigate the above-mentioned research topics. To support sample-efficient learning, we provide high-quality human-teleoperated demonstration data for all tasks. Our extensive experiments present several insightful or even *unexpected* discoveries: sequential finetuning outperforms existing lifelong learning methods in forward transfer, no single visual encoder architecture excels at all types of knowledge transfer, and naive supervised pretraining can hinder agents' performance in the subsequent LLDM.[2]

## 1 Introduction

A longstanding goal in machine learning is to develop a generalist agent that can perform a wide range of tasks. While multitask learning [10] is one approach, it is computationally demanding and not adaptable to ongoing changes. Lifelong learning [65], however, offers a practical solution by amortizing the learning process over the agent's lifespan. Its goal is to leverage prior knowledge to facilitate learning new tasks (forward transfer) and use the newly acquired knowledge to enhance performance on prior tasks (backward transfer).

The main body of the lifelong learning literature has focused on how agents transfer *declarative* knowledge in visual or language tasks, which pertains to *declarative knowledge* about entities and concepts [7, 40]. Yet it is understudied how agents transfer knowledge in decision-making tasks, which involves a mixture of both *declarative* and *procedural* knowledge (knowledge about how to *do* something). Consider a scenario where a robot, initially trained to retrieve juice from a fridge, fails

---

[\*]Equal contribution.
[2]Check the website at https://libero-project.github.io for the code and the datasets.

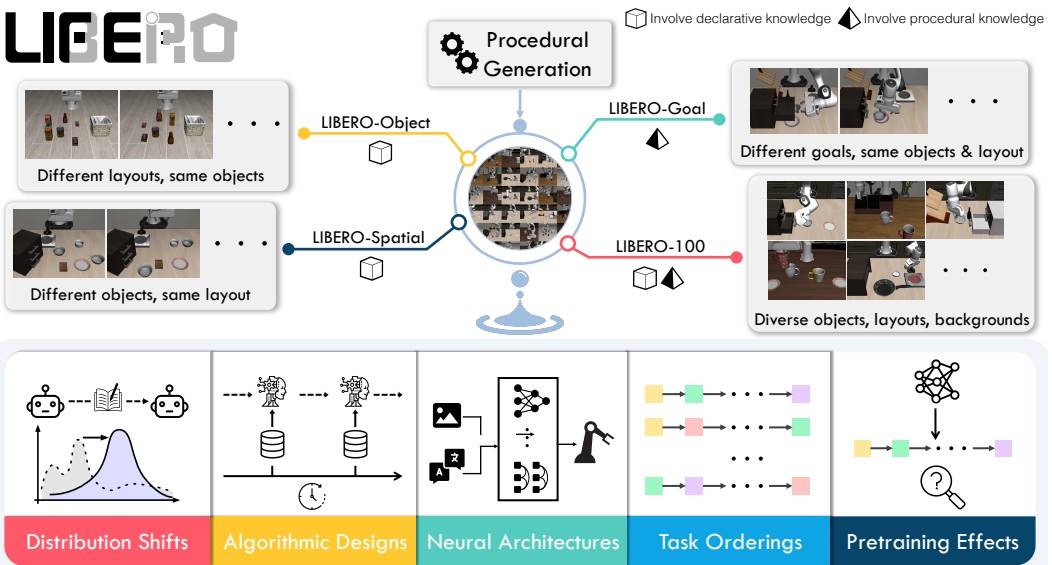

Figure 1: **Top**: LIBERO has four procedurally-generated task suites: LIBERO-SPATIAL, LIBERO-OBJECT, and LIBERO-GOAL have 10 tasks each and require transferring knowledge about spatial relationships, objects, and task goals; LIBERO-100 has 100 tasks and requires the transfer of entangled knowledge. **Bottom**: we investigate five key research topics in LLDM on LIBERO.

after learning new tasks. This could be due to forgetting the juice or fridge's location (declarative knowledge) or how to open the fridge or grasp the juice (procedural knowledge). So far, we lack methods to systematically and quantitatively analyze this complex knowledge transfer.

To bridge this research gap, this paper introduces a new simulation benchmark, LIfelong learning BEchmark on RObot manipulation tasks, LIBERO, to facilitate the systematic study of lifelong learning in decision making (LLDM). An ideal LLDM testbed should enable continuous learning across an expanding set of diverse tasks that share concepts and actions. LIBERO supports this through a procedural generation pipeline for endless task creation, based on robot manipulation tasks with shared visual concepts (declarative knowledge) and interactions (procedural knowledge).

For benchmarking purpose, LIBERO generates 130 language-conditioned robot manipulation tasks inspired by human activities [22] and, grouped into four suites. The four task suites are designed to examine distribution shifts in the object types, the spatial arrangement of objects, the task goals, or the mixture of the previous three (top row of Figure 1). LIBERO is scalable, extendable, and designed explicitly for studying lifelong learning in robot manipulation. To support efficient learning, we provide high-quality, human-teleoperated demonstration data for all 130 tasks.

We present an initial study using LIBERO to investigate five major research topics in LLDM (Figure 1): **1)** knowledge transfer with different types of distribution shift; **2)** neural architecture design; **3)** lifelong learning algorithm design; **4)** robustness of the learner to task ordering; and **5)** how to leverage pre-trained models in LLDM (bottom row of Figure 1). We perform extensive experiments across different policy architectures and different lifelong learning algorithms. Based on our experiments, we make several insightful or even **unexpected** observations:

1. Policy architecture design is as crucial as lifelong learning algorithms. The transformer architecture is better at abstracting temporal information than a recurrent neural network. Vision transformers work well on tasks with rich visual information (e.g., a variety of objects). Convolution networks work well when tasks primarily need procedural knowledge.

2. While the lifelong learning algorithms we evaluated are effective at preventing forgetting, they generally perform *worse* than sequential finetuning in terms of forward transfer.

3. Our experiment shows that using pretrained language embeddings of semantically-rich task descriptions yields performance *no better* than using those of the task IDs.

4. Basic supervised pretraining on a large-scale offline dataset can have a *negative* impact on the learner's downstream performance in LLDM.

## 2 Background

This section introduces the problem formulation and defines key terms used throughout the paper.

### 2.1 Markov Decision Process for Robot Learning

A robot learning problem can be formulated as a finite-horizon Markov Decision Process: $\mathcal{M} = (\mathcal{S}, \mathcal{A}, \mathcal{T}, H, \mu_0, R)$. Here, $\mathcal{S}$ and $\mathcal{A}$ are the state and action spaces of the robot. $\mu_0$ is the initial state distribution, $R : \mathcal{S} \times \mathcal{A} \to \mathbb{R}$ is the reward function, and $\mathcal{T} : \mathcal{S} \times \mathcal{A} \to \mathcal{S}$ is the transition function. In this work, we assume a sparse-reward setting and replace $R$ with a goal predicate $g : \mathcal{S} \to \{0, 1\}$. The robot's objective is to learn a policy $\pi$ that maximizes the expected return: $\max_\pi J(\pi) = \mathbb{E}_{s_t, a_t \sim \pi, \mu_0}[\sum_{t=1}^H g(s_t)]$.

### 2.2 Lifelong Robot Learning Problem

In a *lifelong robot learning problem*, a robot sequentially learns over $K$ tasks $\{T^1, \ldots, T^K\}$ with a single policy $\pi$. We assume $\pi$ is conditioned on the task, i.e., $\pi(\cdot \mid s; T)$. For each task, $T^k \equiv (\mu_0^k, g^k)$ is defined by the initial state distribution $\mu_0^k$ and the goal predicate $g^k$.[3] We assume $\mathcal{S}, \mathcal{A}, \mathcal{T}, H$ are the same for all tasks. Up to the $k$-th task $T^k$, the robot aims to optimize

$$\max_\pi \; J_{\text{LRL}}(\pi) = \frac{1}{k} \sum_{p=1}^k \left[ \mathbb{E}_{s_t^p, a_t^p \sim \pi(\cdot; T^p), \, \mu_0^p} \left[ \sum_{t=1}^L g^p(s_t^p) \right] \right]. \tag{1}$$

An important feature of the lifelong setting is that the agent loses access to the previous $k-1$ tasks when it learns on task $T^k$.

**Lifelong Imitation Learning** Due to the challenge of sparse-reward reinforcement learning, we consider a practical alternative setting where a user would provide a small demonstration dataset for each task in the sequence. Denote $D^k = \{\tau_i^k\}_{i=1}^N$ as $N$ demonstrations for task $T^k$. Each $\tau_i^k = (o_0, a_0, o_1, a_1, \ldots, o_{l^k})$ where $l^k \leq H$. Here, $o_t$ is the robot's sensory input, including the perceptual observation and the information about the robot's joints and gripper. In practice, the observation $o_t$ is often non-Markovian. Therefore, following works in partially observable MDPs [25], we represent $s_t$ by the aggregated history of observations, i.e. $s_t \equiv o_{\leq t} \triangleq (o_0, o_1, \ldots, o_t)$. This results in the *lifelong imitation learning problem* with the same objective as in Eq. (1). But during training, we perform behavioral cloning [4] with the following surrogate objective function:

$$\min_\pi \; J_{\text{BC}}(\pi) = \frac{1}{k} \sum_{p=1}^k \mathbb{E}_{o_t, a_t \sim D^p} \left[ \sum_{t=0}^{l^p} \mathcal{L}\big(\pi(o_{\leq t}; T^p), a_t^p\big) \right], \tag{2}$$

where $\mathcal{L}$ is a supervised learning loss, e.g., the negative log-likelihood loss, and $\pi$ is a Gaussian mixture model. Similarly, we assume $\{D^p : p < k\}$ are not fully available when learning $T^k$.

## 3 Research Topics in LLDM

We outline five major research topics in LLDM that motivate the design of LIBERO and our study.

**(T1) Transfer of Different Types of Knowledge** In order to accomplish a task such as *put the ketchup next to the plate in the basket*, a robot must understand the concept *ketchup*, the location of the *plate/basket*, and how to *put* the ketchup in the basket. Indeed, robot manipulation tasks in general necessitate different types of knowledge, making it hard to determine the cause of failure. We present four task suites in Section 4.2: three task suites for studying the transfer of knowledge about spatial relationships, object concepts, and task goals in a disentangled manner, and one suite for studying the transfer of mixed types of knowledge.

---

[3]Throughout the paper, a superscript/subscript is used to index the task/time step.

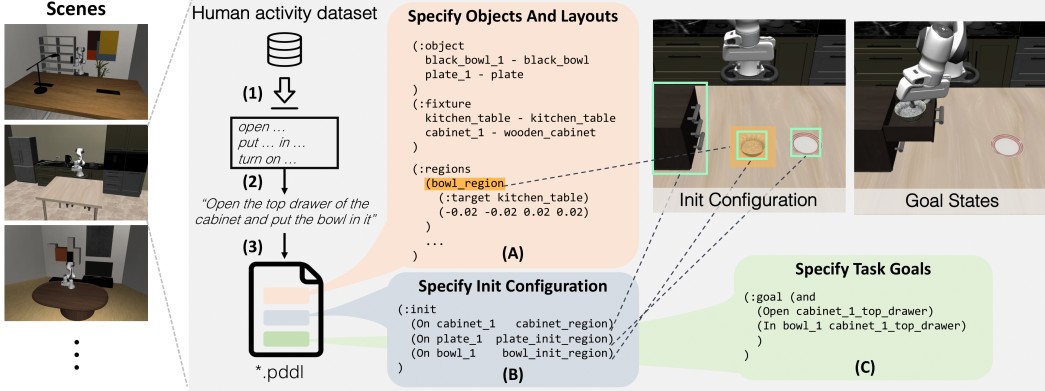

Figure 2: LIBERO's procedural generation pipeline: Extracting behavioral templates from a large-scale human activity dataset (**1**), Ego4D, for generating task instructions (**2**); Based on the task description, selecting the scene and generating the PDDL description file (**3**) that specifies the objects and layouts (**A**), the initial object configurations (**B**), and the task goal (**C**).

**(T2) Neural Architecture Design**   An important research question in LLDM is how to design effective neural architectures to abstract the multi-modal observations (images, language descriptions, and robot states) and transfer only relevant knowledge when learning new tasks.

**(T3) Lifelong Learning Algorithm Design**   Given a policy architecture, it is crucial to determine what learning algorithms to apply for LLDM. Specifically, the sequential nature of LLDM suggests that even minor forgetting over successive steps can potentially lead to a total failure in execution. As such, we consider the design of lifelong learning algorithms to be an open area of research in LLDM.

**(T4) Robustness to Task Ordering**   It is well-known that task curriculum influences policy learning [6, 48]. A robot in the real world, however, often cannot choose which task to encounter first. Therefore, a good lifelong learning algorithm should be robust to different task orderings.

**(T5) Usage of Pretrained Models**   In practice, robots will be most likely pretrained on large datasets in factories before deployment [28]. However, it is not well-understood whether or how pretraining could benefit subsequent LLDM.

## 4   LIBERO

This section introduces the components in LIBERO: the procedural generation pipeline that allows the never-ending creation of tasks (Section 4.1), the four task suites we generate for benchmarking (Section 4.2), five algorithms (Section 4.3), and three neural architectures (Section 4.4).

### 4.1   Procedural Generation of Tasks

Research in LLDM requires a systematic way to create new tasks while maintaining task diversity and relevance to existing tasks. LIBERO procedurally generates new tasks in three steps: **1)** extract behavioral templates from language annotations of human activities and generate sampled tasks described in natural language based on such templates; **2)** specify an initial object distribution given a task description; and **3)** specify task goals using a propositional formula that aligns with the language instructions. Our generation pipeline is built on top of `Robosuite` [76], a modular manipulation simulator that offers seamless integration. Figure 2 illustrates an example of task creation using this pipeline, and each component is expanded upon below.

**Behavioral Templates and Instruction Generation**   Human activities serve as a fertile source of tasks that can inspire and generate a vast number of manipulation tasks. We choose a large-scale activity dataset, Ego4D [22], which includes a large variety of everyday activities with language annotations. We pre-process the dataset by extracting the language descriptions and then summarize them into a large set of commonly used language templates. After this pre-processing step, we use the templates and select objects available in the simulator to generate a set of task descriptions in the

form of language instructions. For example, we can generate an instruction "Open the drawer of the cabinet" from the template "Open ...".

**Initial State Distribution ($\mu_0$)**     To specify $\mu_0$, we first sample a scene layout that matches the objects/behaviors in a provided instruction. For instance, a kitchen scene is selected for an instruction *Open the top drawer of the cabinet and put the bowl in it*. Then, the details about $\mu_0$ are generated in the PDDL language [43, 63]. Concretely, $\mu_0$ contains information about object categories and their placement (Figure 2-(**A**)), and their initial status (Figure 2-(**B**)).

**Goal Specifications ($g$)**     Based on $\mu_0$ and the language instruction, we specify the task goal using a conjunction of predicates. Predicates include *unary predicates* that describe the properties of an object, such as Open(X) or TurnOff(X), and *binary predicates* that describe spatial relations between objects, such as On(A, B) or In(A, B). An example of the goal specification using PDDL language can be found in Figure 2-(**C**). The simulation terminates when all predicates are verified true.

## 4.2 Task Suites

While the pipeline in Section 4.1 supports the generation of an unlimited number of tasks, we offer fixed sets of tasks for benchmarking purposes. LIBERO has four task suites: LIBERO-SPATIAL, LIBERO-OBJECT, LIBERO-GOAL, and LIBERO-100. The first three task suites are curated to disentangle the transfer of *declarative* and *procedural* knowledge (as mentioned in (T1)), while LIBERO-100 is a suite of 100 tasks with entangled knowledge transfer.

**LIBERO-X**     LIBERO-SPATIAL, LIBERO-OBJECT, and LIBERO-GOAL all have 10 tasks[4] and are designed to investigate the controlled transfer of knowledge about spatial information (declarative), objects (declarative), and task goals (procedural). Specifically, all tasks in LIBERO-SPATIAL request the robot to place a bowl, among the same set of objects, on a plate. But there are two identical bowls that differ only in their location or spatial relationship to other objects. Hence, to successfully complete LIBERO-SPATIAL, the robot needs to continually learn and memorize new spatial relationships. All tasks in LIBERO-OBJECT request the robot to pick-place a unique object. Hence, to accomplish LIBERO-OBJECT, the robot needs to continually learn and memorize new object types. All tasks in LIBERO-GOAL share the same objects with fixed spatial relationships but differ only in the task goal. Hence, to accomplish LIBERO-GOAL, the robot needs to continually learn new knowledge about motions and behaviors. More details are in Appendix C.

**LIBERO-100**     LIBERO-100 contains 100 tasks that entail diverse object interactions and versatile motor skills. In this paper, we split LIBERO-100 into 90 short-horizon tasks (LIBERO-90) and 10 long-horizon tasks (LIBERO-LONG). LIBERO-90 serves as the data source for pretraining (**T5**) and LIBERO-LONG for downstream evaluation of lifelong learning algorithms.

## 4.3 Lifelong Learning Algorithms

We implement three representative lifelong learning algorithms to facilitate research in algorithmic design for LLDM. Specifically, we implement Experience Replay (ER) [13], Elastic Weight Consolidation (EWC) [33], and PACKNET [41]. We pick ER, EWC, and PACKNET because they correspond to the memory-based, regularization-based, and dynamic-architecture-based methods for lifelong learning. In addition, prior research [69] has discovered that they are state-of-the-art methods. Besides these three methods, we also implement sequential finetuning (SEQL) and multitask learning (MTL), which serve as a lower bound and upper bound for lifelong learning algorithms, respectively. More details about the algorithms are in Appendix B.1.

## 4.4 Neural Network Architectures

We implement three vision-language policy networks, RESNET-RNN, RESNET-T, and VIT-T, that integrate visual, temporal, and linguistic information for LLDM. Language instructions of tasks are encoded using pretrained BERT embeddings [19]. The RESNET-RNN [42] uses a ResNet as the visual backbone that encodes per-step visual observations and an LSTM as the temporal backbone to process a sequence of encoded visual information. The language instruction is incorporated into the ResNet features using the FiLM method [50] and added to the LSTM inputs, respectively. RESNET-T

---

[4]A suite of 10 tasks is enough to observe catastrophic forgetting while maintaining computation efficiency.

architecture [75] uses a similar ResNet-based visual backbone, but a transformer decoder [66] as the temporal backbone to process outputs from ResNet, which are a temporal sequence of visual tokens. The language embedding is treated as a separate token in inputs to the transformer alongside the visual tokens. The VıT-T architecture [31], which is widely used in visual-language tasks, uses a Vision Transformer (ViT) as the visual backbone and a transformer decoder as the temporal backbone. The language embedding is treated as a separate token in inputs of both ViT and the transformer decoder. All the temporal backbones output a latent vector for every decision-making step. We compute the multi-modal distribution over manipulation actions using a Gaussian-Mixture-Model (GMM) based output head [8, 42, 68]. In the end, a robot executes a policy by sampling a continuous value for end-effector action from the output distribution. Figure 6 visualizes the three architectures.

For all the lifelong learning algorithms and neural architectures, we use behavioral cloning (BC) [4] to train policies for individual tasks (See (2)). BC allows for efficient policy learning such that we can study lifelong learning algorithms with limited computational resources. To train BC, we provide 50 trajectories of high-quality demonstrations for every single task in the generated task suites. The demonstrations are collected by human experts through teleoperation with 3Dconnexion Spacemouse.

## 5 Experiments

Experiments are conducted as an initial study for the five research topics mentioned in Section 3. We first introduce the evaluation metric used in experiments, and present analysis of empirical results in LIBERO. The detailed experimental setup is in Appendix D. Our experiments focus on addressing the following research questions:

**Q1**: How do different architectures/LL algorithms perform under specific distribution shifts?
**Q2**: To what extent does neural architecture impact knowledge transfer in LLDM, and are there any discernible patterns in the specialized capabilities of each architecture?
**Q3**: How do existing algorithms from lifelong supervised learning perform on LLDM tasks?
**Q4**: To what extent does language embedding affect knowledge transfer in LLDM?
**Q5**: How robust are different LL algorithms to task ordering in LLDM?
**Q6**: Can supervised pretraining improve downstream lifelong learning performance in LLDM?

### 5.1 Evaluation Metrics

We report three metrics: FWT (forward transfer) [20], NBT (negative backward transfer), and AUC (area under the success rate curve). All metrics are computed in terms of success rate, as previous literature has shown that the success rate is a more reliable metric than training loss for manipulation policies [42] (Detailed explanation in Appendix E.2). Lower NBT means a policy has better performance in the previously seen tasks, higher FWT means a policy learns faster on a new task, and higher AUC means an overall better performance considering both NBT and FWT. Specifically, denote $c_{i,j,e}$ as the agent's success rate on task $j$ when it learned over $i-1$ previous tasks and has just learned $e$ epochs ($e \in \{0, 5, \ldots, 50\}$) on task $i$. Let $c_{i,i}$ be the best success rate over all evaluated epochs $e$ for the current task $i$ (i.e., $c_{i,i} = \max_e c_{i,i,e}$). Then, we find the earliest epoch $e_i^*$ in which the agent achieves the best performance on task $i$ (i.e., $e_i^* = \arg\min_e c_{i,i,e_i} = c_{i,i}$), and assume for all $e \geq e_i^*$, $c_{i,i,e} = c_{i,i}$.[5] Given a different task $j \neq i$, we define $c_{i,j} = c_{i,j,e_i^*}$. Then the three metrics are defined:

$$\text{FWT} = \sum_{k \in [K]} \frac{\text{FWT}_k}{K}, \quad \text{FWT}_k = \frac{1}{11} \sum_{e \in \{0 \ldots 50\}} c_{k,k,e}$$

$$\text{NBT} = \sum_{k \in [K]} \frac{\text{NBT}_k}{K}, \quad \text{NBT}_k = \frac{1}{K-k} \sum_{\tau=k+1}^{K} \left( c_{k,k} - c_{\tau,k} \right) \tag{3}$$

$$\text{AUC} = \sum_{k \in [K]} \frac{\text{AUC}_k}{K}, \quad \text{AUC}_k = \frac{1}{K-k+1} \left( \text{FWT}_k + \sum_{\tau=k+1}^{K} c_{\tau,k} \right)$$

---

[5]In practice, it's possible that the agent's performance on task $i$ is not monotonically increasing due to the variance of learning. But we keep the best checkpoint among those saved at epochs $\{e\}$ as if the agent stops learning after $e_i^*$.

A visualization of these metrics is provided in Figure 3.

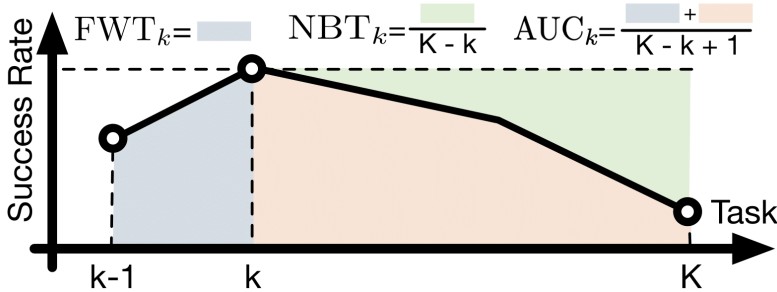

Figure 3: Metrics for LLDM.

## 5.2 Experimental Results

We present empirical results to address the research questions. Please refer to Appendix E.1 for the full results across all algorithms, policy architectures, and task suites.

**Study on the Policy's Neural Architectures (Q1, Q2)** Table 1 reports the agent's lifelong learning performance using the three different neural architectures on the four task suites. Results are reported when ER and PACKNET are used as they demonstrate the best lifelong learning performance across all task suites.

| Policy Arch. | ER | | | PACKNET | | |
|---|---|---|---|---|---|---|
| | FWT($\uparrow$) | NBT($\downarrow$) | AUC($\uparrow$) | FWT($\uparrow$) | NBT($\downarrow$) | AUC($\uparrow$) |
| | | | LIBERO-LONG | | | |
| RESNET-RNN | $0.16 \pm 0.02$ | $\mathbf{0.16} \pm 0.02$ | $0.08 \pm 0.01$ | $0.13 \pm 0.00$ | $0.21 \pm 0.01$ | $0.03 \pm 0.00$ |
| RESNET-T | $\mathbf{0.48} \pm 0.02$ | $0.32 \pm 0.04$ | $\textcolor{purple}{\mathbf{0.32}} \pm 0.01$ | $0.22 \pm 0.01$ | $\textcolor{purple}{\mathbf{0.08}} \pm 0.01$ | $0.25 \pm 0.00$ |
| VIT-T | $0.38 \pm 0.05$ | $0.29 \pm 0.06$ | $0.25 \pm 0.02$ | $\textcolor{purple}{\mathbf{0.36}} \pm 0.01$ | $0.14 \pm 0.01$ | $\textcolor{purple}{\mathbf{0.34}} \pm 0.01$ |
| | | | LIBERO-SPATIAL | | | |
| RESNET-RNN | $0.40 \pm 0.02$ | $0.29 \pm 0.02$ | $0.29 \pm 0.01$ | $0.27 \pm 0.03$ | $0.38 \pm 0.03$ | $0.06 \pm 0.01$ |
| RESNET-T | $\mathbf{0.65} \pm 0.03$ | $\mathbf{0.27} \pm 0.03$ | $\mathbf{0.56} \pm 0.01$ | $0.55 \pm 0.01$ | $\mathbf{0.07} \pm 0.02$ | $\mathbf{0.63} \pm 0.00$ |
| VIT-T | $0.63 \pm 0.01$ | $0.29 \pm 0.02$ | $0.50 \pm 0.02$ | $\mathbf{0.57} \pm 0.04$ | $0.15 \pm 0.00$ | $0.59 \pm 0.03$ |
| | | | LIBERO-OBJECT | | | |
| RESNET-RNN | $0.30 \pm 0.01$ | $\mathbf{0.27} \pm 0.05$ | $0.17 \pm 0.05$ | $0.29 \pm 0.02$ | $0.35 \pm 0.02$ | $0.13 \pm 0.01$ |
| RESNET-T | $0.67 \pm 0.07$ | $0.43 \pm 0.04$ | $0.44 \pm 0.06$ | $\mathbf{0.60} \pm 0.07$ | $\mathbf{0.17} \pm 0.05$ | $\mathbf{0.60} \pm 0.05$ |
| VIT-T | $\mathbf{0.70} \pm 0.02$ | $0.28 \pm 0.01$ | $\mathbf{0.57} \pm 0.01$ | $0.58 \pm 0.03$ | $0.18 \pm 0.02$ | $0.56 \pm 0.04$ |
| | | | LIBERO-GOAL | | | |
| RESNET-RNN | $0.41 \pm 0.00$ | $0.35 \pm 0.01$ | $0.26 \pm 0.01$ | $0.32 \pm 0.03$ | $0.37 \pm 0.04$ | $0.11 \pm 0.01$ |
| RESNET-T | $\textcolor{purple}{\mathbf{0.64}} \pm 0.01$ | $\mathbf{0.34} \pm 0.02$ | $\textcolor{purple}{\mathbf{0.49}} \pm 0.02$ | $0.63 \pm 0.02$ | $\mathbf{0.06} \pm 0.01$ | $0.75 \pm 0.01$ |
| VIT-T | $0.57 \pm 0.00$ | $0.40 \pm 0.02$ | $0.38 \pm 0.01$ | $\mathbf{0.69} \pm 0.02$ | $0.08 \pm 0.01$ | $\mathbf{0.76} \pm 0.02$ |

Table 1: Performance of the three neural architectures using ER and PACKNET on the four task suites. Results are averaged over three seeds and we report the mean and standard error. The best performance is **bolded**, and colored in **purple** if the improvement is statistically significant over other neural architectures, when a two-tailed, Student's t-test under equal sample sizes and unequal variance is applied with a $p$-value of 0.05.

*Findings:* First, we observe that RESNET-T and VIT-T work much better than RESNET-RNN on average, indicating that using a transformer on the "temporal" level could be a better option than using an RNN model. Second, the performance difference among different architectures depends on the underlying lifelong learning algorithm. If PACKNET (a dynamic architecture approach) is

used, we observe no significant performance difference between RESNET-T and VIT-T except on the LIBERO-LONG task suite where VIT-T performs much better than RESNET-T. In contrast, if ER is used, we observe that RESNET-T performs better than VIT-T on all task suites except LIBERO-OBJECT. This potentially indicates that the ViT architecture is better at processing visual information with more object varieties than the ResNet architecture when the network capacity is sufficiently large (See the MTL results in Table 8 on LIBERO-OBJECT as the supporting evidence). The above findings shed light on how one can improve architecture design for better processing of spatial and temporal information in LLDM.

**Study on Lifelong Learning Algorithms (Q1, Q3)**   Table 2 reports the lifelong learning performance of the three lifelong learning algorithms, together with the SEQL and MTL baselines. All experiments use the same RESNET-T architecture as it performs the best across all policy architectures.

| Lifelong Algo. | FWT(↑) | NBT(↓) | AUC(↑) | FWT(↑) | NBT(↓) | AUC(↑) |
|---|---|---|---|---|---|---|
| | LIBERO-LONG | | | LIBERO-SPATIAL | | |
| SEQL | **0.54** ± 0.01 | 0.63 ± 0.01 | 0.15 ± 0.00 | **0.72** ± 0.01 | 0.81 ± 0.01 | 0.20 ± 0.01 |
| ER | 0.48 ± 0.02 | 0.32 ± 0.04 | **0.32** ± 0.01 | 0.65 ± 0.03 | 0.27 ± 0.03 | 0.56 ± 0.01 |
| EWC | 0.13 ± 0.02 | 0.22 ± 0.03 | 0.02 ± 0.00 | 0.23 ± 0.01 | 0.33 ± 0.01 | 0.06 ± 0.01 |
| PACKNET | 0.22 ± 0.01 | **0.08** ± 0.01 | 0.25 ± 0.00 | 0.55 ± 0.01 | **0.07** ± 0.02 | **0.63** ± 0.00 |
| MTL | | | 0.48 ± 0.01 | | | 0.83 ± 0.00 |
| | LIBERO-OBJECT | | | LIBERO-GOAL | | |
| SEQL | **0.78** ± 0.04 | 0.76 ± 0.04 | 0.26 ± 0.02 | **0.77** ± 0.01 | 0.82 ± 0.01 | 0.22 ± 0.00 |
| ER | 0.67 ± 0.07 | 0.43 ± 0.04 | 0.44 ± 0.06 | 0.64 ± 0.01 | 0.34 ± 0.02 | 0.49 ± 0.02 |
| EWC | 0.56 ± 0.03 | 0.69 ± 0.02 | 0.16 ± 0.02 | 0.32 ± 0.02 | 0.48 ± 0.03 | 0.06 ± 0.00 |
| PACKNET | 0.60 ± 0.07 | **0.17** ± 0.05 | **0.60** ± 0.05 | 0.63 ± 0.02 | **0.06** ± 0.01 | **0.75** ± 0.01 |
| MTL | | | 0.54 ± 0.02 | | | 0.80 ± 0.01 |

Table 2: Performance of three lifelong algorithms and the SEQL and MTL baselines on the four task suites, where the policy is fixed to be RESNET-T. Results are averaged over three seeds and we report the mean and standard error. The best performance is **bolded**, and colored in **purple** if the improvement is statistically significant over other algorithms, when a two-tailed, Student's t-test under equal sample sizes and unequal variance is applied with a $p$-value of 0.05.

*Findings:* We observed a series of interesting findings that could potentially benefit future research on algorithm design for LLDM: **1)** SEQL shows the best FWT over all task suites. This is surprising since it indicates all lifelong learning algorithms we consider actually hurt forward transfer; **2)** PACKNET outperforms other lifelong learning algorithms on LIBERO-X but is outperformed by ER significantly on LIBERO-LONG, mainly because of low forward transfer. This confirms that the dynamic architecture approach is good at preventing forgetting. But since PACKNET splits the network into different sub-networks, the essential capacity of the network for learning any individual task is smaller. Therefore, we conjecture that PACKNET is not rich enough to learn on LIBERO-LONG; **3)** EWC works worse than SEQL, showing that the regularization on the loss term can actually impede the agent's performance on LLDM problems (See Appendix E.2); and **4)** ER, the rehearsal method, is robust across all task suites.

**Study on Language Embeddings as the Task Identifier (Q4)**   To investigate to what extent language embedding play a role in LLDM, we compare the performance of the same lifelong learner using four different pretrained language embeddings. Namely, we choose BERT [19], CLIP [52], GPT-2 [53] and the Task-ID embedding. Task-ID embeddings are produced by feeding a string such as "Task 5" into a pretrained BERT model.

*Findings:* From Table 3, we observe *no* statistically significant difference among various language embeddings, including the Task-ID embedding. This, we believe, is due to sentence embeddings functioning as bag-of-words that differentiates different tasks. This insight calls for better language encoding to harness the semantic information in task descriptions. Despite the similar performance, we opt for BERT embeddings as our default task embedding.

| Embedding Type | Dimension | FWT(↑) | NBT(↓) | AUC(↑) |
|---|---|---|---|---|
| BERT | 768 | $0.48 \pm 0.02$ | $\mathbf{0.32} \pm 0.04$ | $0.32 \pm 0.01$ |
| CLIP | 512 | $\mathbf{0.52} \pm 0.00$ | $0.34 \pm 0.01$ | $\mathbf{0.35} \pm 0.01$ |
| GPT-2 | 768 | $0.46 \pm 0.01$ | $0.34 \pm 0.02$ | $0.30 \pm 0.01$ |
| Task-ID | 768 | $0.50 \pm 0.01$ | $0.37 \pm 0.01$ | $0.33 \pm 0.01$ |

Table 3: Performance of a lifelong learner using four different language embeddings on LIBERO-LONG, where we fix the policy architecture to RESNET-T and the lifelong learning algorithm to ER. The Task-ID embeddings are retrieved by feeding "Task + ID" into a pretrained BERT model. Results are averaged over three seeds and we report the mean and standard error. The best performance is **bolded**. No statistically significant difference is observed among the different language embeddings.

**Study on task ordering (Q5)**    Figure 4 shows the result of the study on **Q4**. For all experiments in this study, we used RESNET-T as the neural architecture and evaluated both ER and PACKNET. As the figure illustrates, the performance of both algorithms varies across different task orderings. This finding highlights an important direction for future research: developing algorithms or architectures that are robust to varying task orderings.

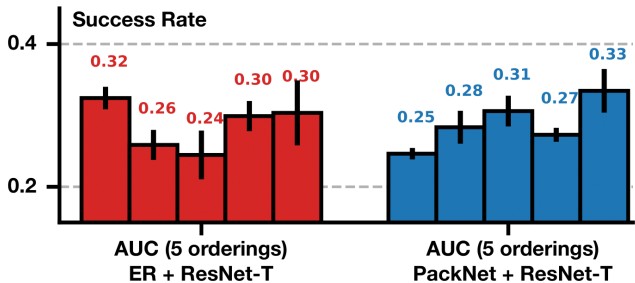

Figure 4: Performance of ER and PACKNET using RESNET-T on five different task orderings. An error bar shows the performance standard deviation for a fixed ordering.

*Findings:* From Figure 4, we observe that indeed different task ordering could result in very different performances for the same algorithm. Specifically, such difference is statistically significant for PACKNET.

**Study on How Pretraining Affects Downstream LLDM (Q6)**    Fig 5 reports the results on LIBERO-LONG of five combinations of algorithms and policy architectures, when the underlying model is pretrained on the 90 short-horizion tasks in LIBERO-100 or learned from scratch. For pretraining, we apply behavioral cloning on the 90 tasks using the three policy architectures for 50 epochs. We save a checkpoint every 5 epochs of training and then pick the checkpoint for each architecture that has the best performance as the pretrained model for downstream LLDM.

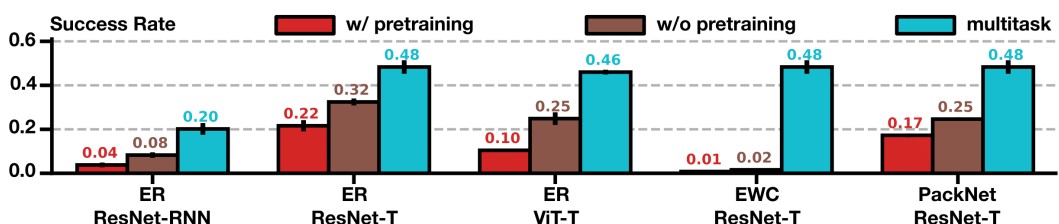

Figure 5: Performance of different combinations of algorithms and architectures without pretraining or with pretraining. The multi-task learning performance is also included for reference.

*Findings:* We observe that the basic supervised pretraining can *hurt* the model's downstream lifelong learning performance. This, together with the results seen in Table 2 (e.g., naive sequential fine-tuning

has better forward transfer than when lifelong learning algorithms are applied), indicates that better pretraining techniques are needed.

**Attention Visualization:**   To better understand what type of knowledge the agent forgets during the lifelong learning process, we visualize the agent's attention map on each observed image input. The visualized saliency maps and the discussion can be found in Appendix E.4.

## 6   Related Work

This section provides an overview of existing benchmarks for lifelong learning and robot learning. We refer the reader to Appendix B.1 for a detailed review of lifelong learning algorithms.

**Lifelong Learning Benchmarks**   Pioneering work has adapted standard vision or language datasets for studying LL. This line of work includes image classification datasets like MNIST [18], CIFAR [34], and ImageNet [17]; segmentation datasets like Core50 [38]; and natural language understanding datasets like GLUE [67] and SuperGLUE [59]. Besides supervised learning datasets, video game benchmarks (e.g., Atari [46], XLand [64], and VisDoom [30]) in reinforcement learning (RL) have also been used for studying LL. However, LL in standard supervised learning does not involve procedural knowledge transfer, while RL problems in games do not represent human activities. ContinualWorld [69] modifies the 50 manipulation tasks in MetaWorld for LL. CORA [51] builds four lifelong RL benchmarks based on Atari, Procgen [15], MiniHack [58], and ALFRED [62]. F-SIOL-310 [3] and OpenLORIS [61] are challenging real-world lifelong object learning datasets that are captured from robotic vision systems. Prior works have also analyzed different components in a LL agent [45, 70, 21], but they do not focus on robot manipulation problems.

**Robot Learning Benchmarks**   A variety of robot learning benchmarks have been proposed to address challenges in meta learning (MetaWorld [73]), causality learning (CausalWorld [1]), multi-task learning [27, 35], policy generalization to unseen objects [47, 24], and compositional learning [44]. Compared to existing benchmarks in lifelong learning and robot learning, the task suites in LIBERO are curated to address the research topics of LLDM. The benchmark includes a large number of tasks based on everyday human activities that feature rich interactive behaviors with a diverse range of objects. Additionally, the tasks in LIBERO are procedurally generated, making the benchmark scalable and adaptable. Moreover, the provided high-quality human demonstration dataset in LIBERO supports and encourages learning efficiency.

## 7   Conclusion and Limitations

This paper introduces LIBERO, a new benchmark in the robot manipulation domain for supporting research in LLDM. LIBERO includes a procedural generation pipeline that can create an infinite number of manipulation tasks in the simulator. We use this pipeline to create 130 standardized tasks and conduct a comprehensive set of experiments on policy and algorithm designs. The empirical results suggest several future research directions: 1) how to design a better neural architecture to better process spatial information or temporal information; 2) how to design a better algorithm to improve forward transfer ability; and 3) how to use pretraining to help improve lifelong learning performance. In the short term, we do not envision any negative societal impacts triggered by LIBERO. But as the lifelong learner mainly learns from humans, studying how to preserve user privacy within LLDM [36] is crucial in the long run.

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
