# OpenReview forum: "LIBERO: Benchmarking Knowledge Transfer for Lifelong Robot Learning"
_NeurIPS.cc/2023/Track/Datasets_and_Benchmarks — NeurIPS 2023 Datasets and Benchmarks Poster_

### Official Review · Reviewer_5K6Q · 2023-07-17
**Libero: a lifelong learning benchmark for robot manipulation tasks**

**Rating:** 6
**Confidence:** 4

**Strengths:**

+ the problem setup of libero is novel and meaningful: previous works in this direction mainly assume multi-task learning, exploring lifelong learning in the context of language-conditioned robot manipulation is an interesting direction for the community

+ it seems the benchmark contains a variety of realistic indoor scenes and household objects

+ perform extensive experiments to examine the effects of multiple factors critical to the performance

+ the supporting documents provide additional experiment details and experiment results to ensure reproducibility; the codebase also seems to be solid and sufficiently documented




**Additional Feedback:**

N/A

**Clarity:**

The paper is overall well written and easy to read.

One thing that is unclear to me is the total number of demonstrations provided - in the paper it states 50 demonstrations for each task -> 130*50=6500 demonstrations in total. On the project website, I'm seeing "65,000 high-quality demonstrations for sample-efficient leanring".

**Correctness:**

In general, the evaluation metrics are sound and the experiments are properly designed .

- Since Libero-100 is supposed to be a comprehensive task suites designed to examine a combination of all 3 distribution shift factors, it is  surprising to see that only 10 tasks are used for evaluation. It is not clear how much of the task space they can cover.

- Regarding the language embedding experiments, since the CLIP embedding seems to perform better than the task id one consistently across three metrics, I would suggest running more trials (20+) and check for statistical significance.


**Documentation:**

The codebase seems to be well documented and the dataset are available. The only issue is that I was not able to download the model checkpoints.

**Limitations:**

- one limitation is the variety of tasks - it seems only pick-and-place and opening drawer tasks are involved in the benchmark. It is not clear whether the findings can be applied to other tasks, e.g. manipulating deformable objects, pouring liquid into a container

- since the benchmark involves realistic indoor scenes and household objects, it would be great to see some real-robot experiments

- collecting human demonstrations is known to be expensive, and the benchmark only provides 50 demonstrations for each task - it is not clear whether these data are sufficient to learn a good policy, especially for the long horizon tasks

**Opportunities For Improvement:**

Many details regarding the simulator, task designs, action space, and data are missing, making it difficult to evaluate the true contribution of the work:

- I was not able to find the underlying simulator the benchmark is built upon, some information regarding its speed would also be quite useful for RL use cases

- more details on the task design are needed: e.g. optimal number of steps to complete each task, number of objects and object types involved, how the 100 tasks in LIBERO-100 are designed, how many scenes are available,

- more details on the action space and observation space are necesary: e.g. for action space what is the form of end-effector control, how the perceptual observation is collected, for sensory input what "information about the robot’s  joints and gripper" are given to the baseline models

- more details on the human demonstration data, e.g. how they are collected, who are the participants, plus some statistics

**Relation To Prior Work:**

The authors are recommended to discuss the following works:

Zeng, A., Florence, P., Tompson, J., Welker, S., Chien, J., Attarian, M., ... & Lee, J. (2021, October). Transporter networks: Rearranging the visual world for robotic manipulation. In Conference on Robot Learning (pp. 726-747). PMLR.

Mees, O., Hermann, L., Rosete-Beas, E., & Burgard, W. (2022). Calvin: A benchmark for language-conditioned policy learning for long-horizon robot manipulation tasks. IEEE Robotics and Automation Letters, 7(3), 7327-7334.



**Summary And Contributions:**

This paper presents Libero, a lifelong learning benchmark that contains 130 language-conditioned robot manipulation tasks to test distribution shift in task goals, object types and locations.  The authors conduct extensive experiments and ablation study on the benchmark to investigate the effects of network architecture, lifelong learning algorithm, language embedding, task ordering and pretraining on performance.

---

> ### Author Response · Authors · 2023-08-20
> **Response to Reviewer**
>
> **Q1: I was not able to find the underlying simulator the benchmark is built upon, some information regarding its speed would also be quite useful for RL use cases.**
>
> LIBERO is built on RoboSuite [1], which is based on the Mujoco simulator.
>
> [1] Robosuite: A modular simulation framework and benchmark for robot learning.
>
> **Q2: More details on the task design are needed: e.g. optimal number of steps to complete each task, number of objects and object types involved, how the 100 tasks in LIBERO-100 are designed, how many scenes are available**
>
> We will include a more detailed description of each task in the Appendix. In our codebase, one can easily check how many objects and what object types are involved. One can also easily compute the average number of steps in expert demonstrations. For the 100 tasks, we have 10 kitchen scenes, 6 living room scenes, and 4 study room scenes.
>
> **Q3: More details on the action space and observation space are necessary**
>
> For action space, we use the 6D pose (position and orientation) of the end effector and a binary gripper command (open/close) to control the robot. For observation space, we get the input images from the front-view camera and the eye-in-hand camera at every timestep. The sensory input about the robot’s joints and gripper is a 7-dim vector of joint angles (6-dim) and gripper openness (1-dim).
>
> **Q4: more details on the human demonstration data, e.g. how they are collected, who are the participants, plus some statistics**
>
> Human demonstrations are collected by the authors using a 3Dconnexion spacemouse (See line 195).
>
> **Q5: Limitation on the variety of tasks - it seems only pick-and-place and opening drawer tasks are involved in the benchmark. It is not clear whether the findings can be applied to other tasks, e.g. manipulating deformable objects, pouring liquid into a container**
>
> Thanks for your suggestion. Note that besides pick-and-place tasks, we also have other contact-rich manipulation tasks such as sliding a plate to a position, and interacting with different kinds of articulated objects. We cover contact-rich interactions that include both prehensile and non-prehensile motions, as well as articulated object manipulation. Adding manipulation tasks with deformable objects is on our TODO list.
>
> **Q6: since the benchmark involves realistic indoor scenes and household objects, it would be great to see some real-robot experiments**
>
> We agree with the reviewer that real-robot experiments will connect what we find to the real-world scenario. The main difficulty of having real-robot experiments lies in the execution length of such experiments due to the sequential nature of LLDM, let alone we need to experiment with different architectures and algorithms.
>
> **Q7: Collecting human demonstrations is known to be expensive, and the benchmark only provides 50 demonstrations for each task - it is not clear whether these data are sufficient to learn a good policy, especially for the long horizon tasks**
>
> Thanks for your question! From Figure 11~13 of the paper, we can see that, with sequential finetuning, the policy can achieve ≥90% success rates for short-horizon tasks (LIBERO-O, LIBERO-S, and LIBERO-G) and ≥60% success rates for long-horizon tasks (LIBERO-Long). In addition, in practice, ideally, a human user will not need to provide more than a few (or even 1) demonstrations for real lifelong learning. We keep the dataset sufficient for learning but also want to make the problem challenging enough to simulate the real-world scenario.
>
> **Q8: Since Libero-100 is supposed to be a comprehensive task suite designed to examine a combination of all 3 distribution shift factors, it is surprising to see that only 10 tasks are used for evaluation. It is not clear how much of the task space they can cover.**
>
> Thanks for your question! 10 tasks are about the upper limit of the current mainstream LL algorithms on robot manipulation tasks [1,2], thus in this paper, we only select 10 tasks for each suite.
>
> [1] Continual world: A robotic benchmark for continual reinforcement learning.
> [2] Lifelong robotic reinforcement learning by retaining experiences.
>
> **Q9: 130*50=6500 demonstrations in total while 65000 on the project website.*
>
> This is a typo and thanks for catching that. It should be 6500.
>
> **Q10: The authors are recommended to discuss the following works: Transporter Networks and Calvin**
>
> Thanks for your reference. Transporter Networks and Calvin also provide diverse manipulation tasks. However, LIBERO focuses on LLDM and tasks are partitioned to facilitate LLDM study. By contrast, it is hard to re-organize tasks from these two works to study different knowledge transfer processes in LLDM.
>
> **Q11: The codebase seems to be well documented and the dataset is available. The only issue is that I was not able to download the model checkpoints.**
>
> The checkpoints are available [here](https://utexas.box.com/s/9s94armp0mjyn2jk71e2adjvj3ub3pb3).

---

> > ### Comment · Reviewer_5K6Q · 2023-08-29
> >
> > I'd like to thank the authors for their response to my and other reviews' comments. The updated manuscript greatly improves the clarity over the original manuscript and provide essential information on the benchmark and experiment details. Nevertheless, some concerns remain:
> >
> > - For Libero-100, only having 10 tasks still seem to be limited considering all the combinations of distribution shift factors we want to examine. On the other hand, looking at table 1 and table 2, the model performance on the 10 tasks in LIBERO-LONG is actually not bad. It would be great if the benchmark can be more challenging and features more diverse tasks.
> >
> > - Details and statistics of the human demonstrations are still missing, e.g. the data collection process, the total number of frames, the distributions of frames within each task suite.

---

> > > ### Author Response · Authors · 2023-08-31
> > > **Follow-up response to Reviewer 5K6Q**
> > >
> > > We would like to thank the reviewer for comments. Here is our follow-up response to the reviewer's remaining concerns:
> > >
> > > **Q1: For Libero-100, only having 10 tasks still seem to be limited considering all the combinations of distribution shift factors we want to examine. On the other hand, looking at table 1 and table 2, the model performance on the 10 tasks in LIBERO-LONG is actually not bad. It would be great if the benchmark can be more challenging and features more diverse tasks.?**
> > >
> > > In principle, all 100 tasks of LIBERO-100 can be used for studying lifelong learning with various distribution shift factors present. We want to note that experiments of 10 tasks are also at the upper bound of computation resources that an average research lab could possibility obtain, thus we use 10 tasks as a standard number of tasks for experiments.
> > > That being said, we do have plans to further extend the task suites to more challenging tasks, which can be easily done through our proposed pipeline of procedural generation.
> > > As for “looking at table 1 and table 2, the model performance on the 10 tasks in LIBERO-LONG is actually not bad”, we believe the reviewer refers to the low NBT metric of the PackNet. We would like to point out that PackNet has worse performance in FWT compared to other task suites.
> > >
> > > ** Q2: Details and statistics of the human demonstrations are still missing, e.g. the data collection process, the total number of frames, the distributions of frames within each task suite**
> > >
> > > Human demonstrations are collected by the authors using a 3Dconnexion space mouse (See line 195). The data collection process follows the [teleoperation pipeline](https://robosuite.ai/docs/algorithms/demonstrations.html) in robosuite. As for the demonstrator, we had two expert demonstrators to collect the data.
> > > We have also provided the demonstration collection script at [here](https://github.com/Lifelong-Robot-Learning/LIBERO/blob/master/scripts/libero_100_collect_demonstrations.py). We provide the detailed distributions of frames at [here](https://libero-project.github.io/datasets).

---

### Official Review · Reviewer_2kT5 · 2023-07-20
**Good benchmark and offer good insights**

**Rating:** 6
**Confidence:** 4
**Correctness:** Yes
**Clarity:** YES

**Strengths:**

1. The paper is generally well-written.
2. There is a need to build a robotics benchmark for lifelong learning and the authors construct the benchmark in a reasonable manner.
3. Offers good analysis and identifies common issues for the current paradigm.
4.  The paper supply baselines cover different training strategies and different network structures, which are a good starting point to improve upon.


**Additional Feedback:**

Some robotics work like the widely used CLIPort reports the possibility of transferring to unseen objects or even unseen colors through the use of CLIP. However, this paper shows language embeddings are not useful.  Can you offer some thoughts on this? CLIPort is trained under the multi-task setting using a transporter network. is it possible the model used in this work is not able to leverage language embeddings as CLIPort did? Or maybe this is because only models trained under multi-task settings are able to leverage language embeddings.

**Documentation:**

N/A

**Opportunities For Improvement:**

1. The paper uses template-generated language which limits linguistic diversity. Maybe, using a paraphrase model or LLMs can help.
2. in the initial states generation process, when authors sample low-level object locations?  How do you avoid bias in this sampling process?
3. This point is minor: a proof of concept on real robots would be good as the goal is to align with real robot learning.
4. A table of comparison with other benchmarks would help the readers understand the contributions better.
5. For robot control, are the authors using position control or velocity control, or maybe just the end-effector control through IK? It's unclear what is the control input/output and what is the output of learning algorithms. The control type might also have some impacts on the learning outcome, and it seems this is not discussed.

**Relation To Prior Work:**

Yes

**Summary And Contributions:**

The authors proposed a benchmark for lifelong robot learning. In this benchmark, in addition to constructing a dataset to systematically study lifelong learning, authors try to answer some key questions regards to lifelong robot learning and offer some valuable insights to the community. I really appreciate these efforts.

---

> ### Author Response · Authors · 2023-08-20
> **Response to Reviewer**
>
> **Q1: The paper uses template-generated language which limits linguistic diversity. Maybe, using a paraphrase model or LLMs can help.**
>
> Thanks for your suggestions! Yes, using LLM to generate task descriptions is a very good idea. We will explore this idea in future versions of LIBERO.
>
> **Q2: In the initial states generation process, when authors sample low-level object locations? How do you avoid bias in this sampling process?**
>
> We first define the layout (which includes the location ranges of objects) in the PDDL descriptions of tasks and then sample specific locations at the initialized step of each task, uniformly at random.
>
> **Q3: This point is minor: a proof of concept on real robots would be good as the goal is to align with real robot learning.**
>
> Thanks for your suggestions! We will definitely try to do real robot experiments in the future. The major difficulty is the task execution length as in lifelong learning we need to learn and evaluate task performance sequentially, which makes the real robot experiment particularly hard.
>
> **Q4: A  table of comparison with other benchmarks would help the readers understand the contributions better.**
>
> We summarize the comparison of LIBERO against prior benchmarks in the table in the common response.
>
> **Q5: For robot control, are the authors using position control or velocity control, or maybe just the end-effector control through IK? It's unclear what is the control input/output and what is the output of learning algorithms. The control type might also have some impacts on the learning outcome, and it seems this is not discussed.**
>
> We use the 6D pose (position and orientation) of the end effector and one dimension of the gripper openness to control the robot. This is the same setting adopted in prior works [1]. But we agree that different control input/output might have an impact and it is worth investigating how they influence the LLDM performance in the future.
>
> [1] What Matters in Learning from Offline Human Demonstrations for Robot Manipulation.
>
> **Q6: Some robotics work like the widely used CLIPort reports the possibility of transferring to unseen objects or even unseen colors through the use of CLIP. However, this paper shows language embeddings are not useful. Can you offer some thoughts on this? CLIPort is trained under the multi-task setting using a transporter network. is it possible the model used in this work is not able to leverage language embeddings as CLIPort did? Or maybe this is because only models trained under multi-task settings are able to leverage language embeddings.**
>
> A: Thanks for your question! First, we agree with the reviewer that lifelong learning is a more challenging problem than multitask learning (MTL). Note that in the computer vision domain, we see that lifelong learners struggle to even learn classification problems online. Secondly, in the original CLIPort paper, it is mentioned in Sec. 4.5 that "Tasks that require generalizing to novel colors, shapes, and objects are more difficult and all our agents achieve relatively lower performance on these tasks, as shown in Figure 3". Hence, we think CLIPort shows some positive forward transfer to unseen concepts but is not significant enough to say it can leverage language to achieve zero-shot or few-shot generalization to those concepts. Lastly, we think both the objects and motion behaviors in LIBERO are more diverse and more difficult for an agent to learn.

---

> > ### Comment · Reviewer_2kT5 · 2023-08-31
> >
> > Thanks for the thorough and comprehensive response. I appreciate the efforts to address my questions, especially the question about zero-shot generalizations through language.
> >
> > Even though some weaknesses remain, I recommend accepting this paper.

---

### Official Review · Reviewer_kBSE · 2023-07-20
**Good benchmark**

**Rating:** 7
**Confidence:** 4

**Strengths:**

(1) The procedural generation pipeline enables creating a large number of diverse and customizable tasks for studying lifelong robot learning. This is more scalable and flexible than curating tasks manually.

(2) The task suites are thoughtfully designed to disentangle different factors like objects, spatial relationships, goals, etc. This will allow targeted study of various knowledge transfer capabilities.

(3) I have been using the benchmark for quite a while, and the authors seem to be pretty responsible for maintaining it.

**Additional Feedback:**

Can you provide more analysis into why pretraining hurt performance in the lifelong learning setting? Is there something specific to your implementation or the nature of LLDM tasks that causes this negative transfer?

**Clarity:**

Overall this paper is well written; however, the pre-training experiments setting needs to be clarified, and also the results reported in the figure are not very clear. I will suggest the authors also report the pre-training results on the training 90 tasks.

**Correctness:**

The negative impact of pretraining on downstream performance is unintuitive, since prior works show pretraining often improves generalization. The authors should provide more analysis into why pretraining hurt adaptation capability in this case. Is there something specific to lifelong learning or the implementation that causes this?

**Documentation:**

Documentation is good, only some minor typos need to be fixed.

**Limitations:**

Yes

**Opportunities For Improvement:**

1. Only behavioral cloning is used for training, rather than full reinforcement learning. BC is essentially just supervised learning, so it's worth exploring how reinforcement learning algorithms perform on these benchmarks. Using RL could reveal new challenges compared to BC. Have you experimented with reinforcement learning algorithms on the benchmarks? How much more difficult is it to train RL compared to BC? What insights emerge from using RL?

2. The benchmark is currently restricted to lifelong imitation learning, but the multimodal demonstrations could support other research areas beyond lifelong learning. It would be useful to compare LIBERO to other multimodal robotics benchmarks and highlight the unique aspects it provides.

**Relation To Prior Work:**

How does LIBERO compare to existing multimodal language-conditioned robotics benchmarks? What unique capabilities does it offer researchers beyond just lifelong learning?

**Summary And Contributions:**

This paper introduces LIBERO, a new benchmark for studying lifelong learning in robot manipulation tasks. The key contribution is a procedural generation pipeline that can create a large number of diverse tasks by sampling language instructions, objects, spatial relationships, and goals. Four task suites with 130 total tasks are provided to study different types of knowledge transfer. Extensive experiments compare neural architectures and lifelong learning algorithms on these tasks. The results offer several interesting insights, such as transformers outperforming RNNs, and simple finetuning achieving the best forward transfer. Overall, LIBERO seems like a useful benchmark that can facilitate more systematic research on lifelong learning for robotics.

---

> ### Author Response · Authors · 2023-08-20
> **Response to Reviewer**
>
> **Q1: Only behavioral cloning is used for training, rather than full reinforcement learning. BC is essentially just supervised learning, so it's worth exploring how reinforcement learning algorithms perform on these benchmarks. Using RL could reveal new challenges compared to BC. Have you experimented with reinforcement learning algorithms on the benchmarks? How much more difficult is it to train RL compared to BC? What insights emerge from using RL?**
>
> Thanks for your question. There are two main difficulties in performing reinforcement learning on LIBERO tasks. On the one hand, it is difficult to define dense reward functions for diverse procedurally generated tasks. On the other hand, 3D vision-based reinforcement learning on manipulation tasks is difficult even with dense rewards [1,2].
>
> [1] On the Efficacy of 3D Point Cloud Reinforcement Learning.
>
> [2] Maniskill2: A unified benchmark for generalizable manipulation skills
>
> **Q2: The benchmark is currently restricted to lifelong imitation learning, but the multimodal demonstrations could support other research areas beyond lifelong learning. It would be useful to compare LIBERO to other multimodal robotics benchmarks and highlight the unique aspects it provides.**
>
> Thanks for your suggestions! We summarize the comparison of LIBERO against prior benchmarks in the table in the common response.
> Beyond lifelong learning, we hope LIBERO can also help the robotics community in the following topics:
> - **Transfer learning, meta-learning, and multi-task learning**: LIBERO provides well-organized task suites and demonstrations. They are easy to be reorganized into the forms of these three kinds of topics.
> - **Skill discovery**: the simulation environments and multi-task demonstrations provided in LIBERO are good testbeds for skill discovery methods in the robot learning area that discovers (discrete or continuous) skills in an unsupervised way.
> - **Architecture search**: LIBERO has a wide range of manipulation tasks that can be used to perform neural architecture search for robot learning.
> - **Imitation learning and robot learning**: At the very least, LIBERO’s task suites could also benefit fundamental research in imitation learning and robot learning. It is particularly useful for testing the robustness and generalization of a proposed robot learning algorithm.
>
> **Q2: The negative impact of pretraining on downstream performance is unintuitive, since prior works show pretraining often improves generalization. The authors should provide more analysis into why pretraining hurt adaptation capability in this case. Is there something specific to lifelong learning or the implementation that causes this?**
>
> Thanks for your question. The intuition behind this can be that: if the network has been pretrained, the perception network will tend to extract some fixed pattern of the visual inputs, and the policy network will tend to use some fixed ways to leverage these patterns to perform tasks. However, in downstream tasks, the network needs to extract new visual patterns (object and spatial variations) and perform tasks in different ways (spatial and goal variations), which makes the pretrained network worse than a randomly initialized network.
>
> **Q3: Overall this paper is well written; however, the pre-training experiments setting needs to be clarified, and also the results reported in the figure are not very clear. I will suggest the authors also report the pre-training results on the training 90 tasks.**
>
> Thanks for your question. The statistics are provided [here](https://drive.google.com/file/d/17mQisOgIDFYyOd2qCuqB1P8BUnG_54zO/view?usp=sharing). As we can see, many tasks have success rates near zero because of the challenge of learning 90 tasks simultaneously.

---

> > ### Comment · Reviewer_kBSE · 2023-08-30
> >
> > Thanks for the author's detailed response. I really appreciate the provided success rate statistics plots for each individual, could the authors provide the numbers for them? Also, if possible, providing not only the LIBERO-90 multi-task training data, but also other suites like LIBERO-10's and LIBERO-Spatial's. These task-wise statistics would be very helpful for developers as a reference. These statistics could be posted on the website or pointed somewhere in the Github repo.
> >
> > The authors also mentioned some difficulties for training RL in this environments, which make me more curious how does standard RL libraries perform on this benchmark. This will also greatly enhance the contributions of this work in the RL community.

---

### Official Review · Reviewer_XKZ6 · 2023-07-21
**Rigorous work with many interesting future directions**

**Rating:** 7
**Confidence:** 2
**Clarity:** Yes.

**Strengths:**

- The benchmark is thoughtfully designed and flexible/extendable.
- A broad range of baselines are studied and several preliminary findings are presented, suggesting plenty of future research directions.
- The experimental analysis is rigorous. For example, the statistical significance of each result is provided.
- The paper is clearly written and articulate.

**Additional Feedback:**

N/A

**Correctness:**

Yes. The claims are careful and well-supported, the analysis is thorough, and the experimental setup and evaluation methods are clearly described.

**Documentation:**

Yes.
Documentation on how a user can create their own task (assuming they can) using the pipeline would be nice.

**Ethics:**

No.

**Limitations:**

Yes.

**Opportunities For Improvement:**

- Given that the tasks are procedurally generated, I would have loved to see some experiments involving curriculum learning—not just task ordering, but increasing task difficulty within each task type.
- The main focus is on imitiation learning because the sparse reward setting is otherwise too challenging. Since pick-place style robot manipulation tasks are relatively easy to specify dense rewards for, I am curious if the authors explored using a dense reward setting.

**Relation To Prior Work:**

Yes.

**Summary And Contributions:**

LIBERO is a benchmark of lifelong learning in robot manipulation tasks. These tasks can be procedurally generated, making the benchmark scalable and extendable. Four task suites with a total of 130 tasks are included; each of the first three is designed to probe a different type of distribution shift (spatial, object, goal), with the fourth mixing them together and exploring longer horizon tasks. Tasks differ by initial state and goal but share the same overall reward structure, action-space, state-space and transition dynamics. For baselines, three SOTA lifelong learning algorithms are analyzed, along with sequential fine-tuning and multitask learning. Additionally, three vision-language policy networks are analyzed. Human demonstrations are provided.

This paper also presents several preliminary findings using LIBERO: sequential finetuning outperforms lifelong learning methods in forward transfer; different architectures have different strengths and weaknesses when it comes to knowledge transfer; using pretrained language embeddings of task descriptions is not significantly better than using task IDs for task-conditioned policies; and pretraining can hinder performance downstream.

---

> ### Author Response · Authors · 2023-08-20
> **Response to Reviewer**
>
> **Q1: Given that the tasks are procedurally generated, I would have loved to see some experiments involving curriculum learning—not just task ordering, but increasing task difficulty within each task type.**
>
> We appreciate the reviewers’ suggestion and we believe enabling curriculum learning will be a very interesting future direction, which is definitely possible with LIBERO. In fact, we have thought about creating task curriculums automatically. The major concern comes from the definition of task difficulty. To the best of our knowledge, defining task difficulty or task similarity is still a core problem in the robotics community. Task difficulty, at the moment, is more of a subjective evaluation than some quantitative mathematical measure. For instance, it is hard to say whether picking up one object is more difficult than picking up another. It is also not clear whether what is hard for humans is also hard for robots.
>
> Since this work mainly focuses on lifelong learning, we eventually reach the claim that we would like the agent to learn well even under slightly adversarial curricula, because in a real-world scenario, there is no guarantee that the agent will learn in a good curriculum. Therefore, we sample task orders and challenge researchers to find lifelong learning methods that are robust to the task ordering.
>
> **Q2: The main focus is on imitation learning because the sparse reward setting is otherwise too challenging. Since pick-place style robot manipulation tasks are relatively easy to specify dense rewards for, I am curious if the authors explored using a dense reward setting.**
>
> Thanks for your suggestion. Currently, we have not yet explored reinforcement learning (RL) in LLDM. Specifying a dense reward function is known to be hard for manipulation tasks, let alone that we aim for a never-ending sequence of tasks. In addition, single-task vision-based RL is already quite challenging [1,2]. Most of the tasks in LIBERO cannot be easily realized by RL with sparse rewards. But we agree with the reviewer that it would be an important future direction to enable lifelong RL on LIBERO.
>
> [1] On the Efficacy of 3D Point Cloud Reinforcement Learning.
>
> [2] Maniskill2: A unified benchmark for generalizable manipulation skills.
>
> **Q3: Documentation on how a user can create their own task (assuming they can) using the pipeline would be nice.**
>
> We have published documentation [here](https://lifelong-robot-learning.github.io/LIBERO/html/getting_started/overview.html). We will make sure to keep polishing the documentation.

---

### Official Review · Reviewer_3N9d · 2023-07-21
**A nice benchmark with a solid initial documentation and investigation that will hopefully continue to grow over time**

**Rating:** 7
**Confidence:** 4

**Strengths:**

The paper has a lot of strengths, listing some of them:
* It is decently motivated, well structured and written.
* The four task suits that are designed to examine shift in object types, spatial arrangement of objects, the task goals, and their mixtures enable a more principled evaluation and benchmarking of strategies than some prior benchmarks have (that simply concatenate some existing datasets).
* The paper uses the proposed tool to investigate a reasonable amount of interesting questions in an empirically decent first examination.
* The website and documentation are nicely structured for this first version of LIBERO.

**Additional Feedback:**

The paper has used three metrics in their empirical investigation, which already improves upon the naive use of just an average loss or similar (as mentioned in the paper too).
Perhaps it might be worth pointing the reader to e.g. "CLEVA-Compass: A continual learning evolution assessment compass to promote research transparency and comparability" (Mundt et al ICLR 2022), which provides exhaustive summaries of the various set-ups, empirical metrics, and their purpose in lifelong learning, towards more exhaustive future evaluation of LIBERO.

**Clarity:**

LIBERO seems like a very nice and comprehensive environment and the paper is generally very well written and structured. The advantages of LIBERO with respect to existing environments and why it is uniquely suited to answer the highlighted questions in the paper remain a bit unclear. The primary advantage seems to be the partitioning of the created datasets, which is unclear whether this is an advantage of the simulator or the created benchmark.


**Correctness:**

There do not seem to be any incorrect statements in the paper per se. The highlight of “unexpected” observations is perhaps a bit questionable, especially with respect to sentences like “sequential fine tuning is better at forward transfer”.  While important to point out, labelling this as some novel surprising contribution is a bit misleading. The entire discussion on stability-plasticity (sensitive) tradeoff in lifelong learning makes it clear that when limited in terms of information capacity, being able to overwrite existing parameters (i.e. allowing forgetting in fine-tuning) makes it easier to encode new information in contrast to having to striking a balance.
I don’t see this as a major detriment of the paper, but I would recommend being more careful with labelling such statements as surprising or even “unexpected”. The paper really does not need this kind of highlight to shine.

**Documentation:**

The documentation on the website includes instructions on how to install, how code is structured, how algorithms can be called/modified and various  environments/datasets created/used.
I anticipate some of these instructions to grow over times, as they are in parts still very brief and hard to understand for novices, but commend the rather exhaustive nature of the documentation state at this point already.


**Ethics:**

There are no ethical concerns with this work

**Limitations:**

It remains a bit unclear of how challenging it would be to create new environments and generate own datasets. Perhaps this is because I have personally never worked with this kind of simulator before. Perhaps it is due to the descriptions of related work and respective documentation on creating custom datasets (which is just a small paragraph) being too short (see below). In either way, there seems to be room for improvement here.
In similar spirit, it is unclear how easy/feasible the outlined procedure in section 4.1 for procedural generation of tasks is for prospective users to mimic.

**Opportunities For Improvement:**

There are many ways in which the paper could still be improved, but some of these are expected to fall into a future work category. I.e. there could always be a more exhaustive evaluation of algorithms (i.e. more metrics & algos), but the present assessment is also reasonable. More importantly, and as outlined below in the respective sections, the primary improvements for the present version could be with respect to highlighting some limitations in the way environments can be designed, providing some clarification with respect to uniqueness and more detailed relation to other types of similar work (rather than just living them).

**Relation To Prior Work:**

The relation to prior work is very coarsely pointed out, but would definitely benefit from some more detailed explanations. Inferring from context, I believe that the proposed environment is sufficiently different from existing ones through the partitioning into the three LIBERO-X variants. It does however remain unclear what the in-depth differences and commonalities are with respect to e.g. CausalWorld, MetaWorld, ContinualWorld etc.
This should ideally be clarified, e.g. in the appendix.


**Summary And Contributions:**

The paper proposes LIBERO, a novel benchmark and benchmark generator  for lifelong learning for robot manipulation. It is motivated by providing a tool that allows investigating transfer in both declarative and procedural knowledge. The respective procedural generation pipeline has been used to created 4 versions of task suites that allow investigation of shifts in object types, spatial arrangement of objects, task goals, and their mixtures. These are then examined in a preliminary empirical examination to highlight the advantages and the types of scientific questions LIBERO allows to analyze.

---

> ### Author Response · Authors · 2023-08-20
> **Response to Reviewer**
>
> **Q1: More exhaustive evaluation of algorithms. Highlight limitations in the way environments are designed. Point out the relation to similar works**
>
> Thank you very much for your suggestions.
> - **Algorithm:** We agree that a more comprehensive evaluation of existing lifelong algorithms could benefit future research. At present, as evaluating lifelong learning methods is not cheap due to their sequential nature, we choose the most representative algorithm for each of the three major lifelong learning approaches. We also devoted effort to making everything accessible and reproducible so that future research could conveniently include their algorithms into our framework.
> - **Environment:** LIBERO generates task environments in 3 steps: getting language descriptions from human activities -> specifying an initial object distribution -> specifying task goals, so the limitations could be: 1) limited task variety. We choose the everyday activities dataset, Ego4D, for generating task descriptions, so other behaviors and objects that are not in this dataset cannot be generated in LIBERO; 2) We convert every task description into a kitchen scene, which is a desktop single robotic arm manipulation scene. However, a task description in Ego4D can correspond to other kinds of specific scenes; 3)  Our paradigm selects tasks from a fixed set (although initial distributions and goals can be randomized), which means we cannot generate infinite kinds of different task descriptions. Future work can seek to use LLMs or other generation models to first generate (potentially infinite) task descriptions and then use some 3D mesh generation models to generate object instances; 4) It is hard to define “task difficulties” that can be leveraged to make the tasks in each suite strictly the same difficulty (or increasing difficulty). Currently, we just select tasks based on human expertise.
>
> For a more detailed comparison with other benchmarks, please refer to the Table in the common response.
>
> **Q2: How easy is it to create new tasks? Make it clear.**
>
> Thanks for your question. We will make sure that the documentation for creating new tasks has all the necessary details. We think it is reasonably simple for prospective users to use PDDL to generate new tasks. It works like this: 1) collect a set of scene backgrounds, textures, robots, and object meshes -> 2) write the PDDL description file to specify objects, layouts, initial distributions, and goals for each task -> 3) generate tasks in the simulator automatically. Note that the procedural generation of simulation tasks has also been used in prior works [1,2].
>
> [1] Behavior: Benchmark for everyday household activities in virtual, interactive, and ecological environments.
>
> [2] Cora: Benchmarks, baselines, and metrics as a platform for continual reinforcement learning agents.
>
> **Q3: The highlight of “unexpected” observations is a bit questionable.**
>
> We will modify our language to remove the confusion. We originally claimed this phenomenon to be unexpected because the finetuning baseline usually performs much worse than most lifelong learning algorithms in both forward and backward transfer, on continual image classification tasks. But here we observed that representative lifelong learning algorithms fall short on forward transfer on manipulation tasks. This observation might encourage future research to design better algorithms that boost forward transfer.
>
> **Q4: Uniqueness to prior works is a bit vague. The primary advantage seems to be the partitioning of the created datasets, which is unclear whether this is an advantage of the simulator or the created benchmark.**
>
> The comparison against prior benchmarks is summarized in the table in the common response. Yes, the primary advantage of LIBERO can be thought of as the partitioning of the created datasets (or benchmarks), but it is by the author’s efforts rather than the advantage of the simulator. The partitioning relies on our designs of task variations, networks, algorithm selections, task ordering, language descriptions, and the procedural generation pipeline, which are all contributions of this paper.
>
> **Q5: Make sure the documentation and code are ready and clear.**
>
> Thanks for your suggestion. We will definitely keep polishing the codebase since we sincerely hope that this work could largely benefit both the robotics and the lifelong learning communities.
>
> **Q6: Consider more comprehensive evaluation metrics.**
>
> Thanks for pointing this work to us and we have included CLEVA-Compass as a reference to the comprehensive empirical metrics for LLDM. We will definitely consider including more detailed metrics for the evaluation of LIBERO.

---

> > ### Comment · Reviewer_3N9d · 2023-08-28
> > **Acknowledgement of rebuttal**
> >
> > Thank you for the response to address concerns and answer remaining questions.
> >
> > I appreciate the proposed changes and hope to see the paper be improved even further.
> > In any case, looking also at the other reviewers' suggestions, I find my initial assessment confirmed.
> >
> > My rating will thus remain to recommend acceptance of the paper (7).

---

### Author Response · Authors · 2023-08-20
**Common Response to All Reviewers**

We thank all reviewers for their positive opinion and constructive feedback. We address individual questions in individual responses. As many reviewers ask for a comparison of LIBERO against prior benchmarks. We provide the following table that summarizes the difference.

| |ContinualWorld [1] | CORA [2] | CausalWorld [3] | CompoSuite [4] | LIBERO |
|----------|----------|----------|----------|----------|----------|
| Disentangled Knowledge Transfer  | $\times$ | $\times$ | $\surd$ | $\surd$  | $\surd$ |
| Imitation Learning & Demonstrations | $\times$ | $\times$ | $\times$ | $\times$ | $\surd$ |
| Reinforcement Learning| $\surd$ | $\surd$ | $\surd$ | $\surd$ | $\surd$ (sparse reward only) |
| Procedurally Generated Environment | $\times$ | $\surd$ | $\surd$ | $\surd$ | $\surd$ |
| Learning with Language Description | $\times$ | $\times$ | $\times$ | $\surd$ | $\surd$ |
| Policy Architectures | $\times$ | $\times$ | $\times$ | $\times$ | $\surd$ |
| Lifeglong Learning Algorithms | $\surd$ | $\surd$ | $\times$ | $\times$ | $\surd$ |
| Task Ordering | $\surd$ | $\times$ | $\times$ | $\times$ | $\surd$ |
| Pretraining | $\times$ | $\times$ | $\times$ | $\times$ | $\surd$ |

**References:**

[1] Continual World: A Robotic Benchmark For Continual Reinforcement Learning.

[2] CORA: Benchmarks, Baselines, and Metrics as a Platform for Continual Reinforcement Learning Agents.

[3] CausalWorld: A Robotic Manipulation Benchmark for Causal Structure and Transfer Learning.

[4] CompoSuite: A Compositional Reinforcement Learning Benchmark.

---

### Decision · Program_Chairs · 2023-09-22

**Decision:**

Accept (Poster)

**Comment:**

Reviewers agreed that this work was clearly written and presented a high-quality benchmark that has the potential to be useful to the robot manipulation community. Feedback in response to reviewer’s specific questions should be incorporated in any final version of the paper, particularly including the additional table of comparisons, information about the process of creating a new task, and discussions of future work as it pertains to reinforcement learning.

Pros:
+ Benchmark is useful, contains important and realistic scenarios, and expands on existing benchmarks in well-defined ways.
+ Clearly written paper on an important topic.
+ Documentation for benchmark use is clear and well maintained.
+ Benchmark can be expanded with new tasks over time.

Cons:
- Please incorporate comparisons to other benchmarks as per response to reviewers.
- Discussion of, and ideally examples of, full reinforcement learning would strengthen the benchmark.
- The negative impacts of pre-training should be more clearly analyzed and explained.
- Real robot experiments would help support claims of applicability.